# Analysis of Rare Plant Occurrence Data for Monitoring Prioritization

**Hailey Laskey [1],\*, Elizabeth D. Crook [2] and Sarah Kimball [3]** 

[1]   Department of Ecology and Evolutionary Biology, University of California, Irvine, Steinhaus Hall, Irvine, CA 92697, USA

[2]   Department of Earth System Science, University of California, Irvine, Croul Hall, Irvine, CA 92697, USA; e.crook@uci.edu

[3]   Center for Environmental Biology, University of California, Irvine, Steinhaus Hall, Irvine, CA 92697, USA; skimball@uci.edu

\*   Correspondence: hlaskey@uci.edu

**Abstract:** Efforts to conserve rare plant species can be limited by a lack of time and funding for monitoring. Understanding species occurrence and distribution patterns within existing protected habitat and throughout the entire species range can help stewards prioritize rare plant monitoring. We created a database of rare plant occurrences from public, private, and research sources to analyze the distribution of rare plant species throughout the existing protected area within the Nature Reserve of Orange County in California, USA. We analyzed species occurrence relative to the urban edge, roads, trails, and mean high tide line. We also determined the vegetation community with the highest number of rare plant species to help prioritize habitats for conservation and restoration. We found that some parts of protected areas have more rare plant species and we also found sampling biases on the location of occurrence data. We found that rare species occur close to roads and trails and the mean high tide line. Rare species were in all vegetation communities within the reserve, including degraded areas. Using patterns of distribution and considering the immediate threats to a rare species population can help land managers and stewards prioritize monitoring toward the most threatened species.

**Keywords:** rare plants; reserve management; species prioritization; sampling bias; California

## 1. Introduction

Rare plant species have ecological, political, and intrinsic value, leading land managers and conservationists to advocate for their protection. Reserves are established in areas with a high concentration of rare and sensitive listed species [1,2] to help protect the biodiversity contributions of rare species [3]. However, managing reserves with multiple rare species can prove challenging. Evaluating the types of rarity can help identify appropriate actions to protect populations. Rare species can be locally rare but have large geographic ranges, locally abundant with small ranges, or locally rare with large ranges, and can thus have wide or narrow habitat specificity [4]. Rare species may also be rare because of their life history characteristics, such as their dependency on a disturbance regime or a specific pollinator [5–7]. With diverse life histories, species ranges may be at the periphery of protected land, with only one population or maybe even one individual [8]. These species are subject to inbreeding or extinction just from their small population sizes alone [9,10]. It's important that conservation efforts cover the diversity of rare species characteristics when establishing reserves.

Protected areas are increasingly fragmented by urban or agricultural development, which is a barrier for rare species movement and leads to anthropogenic threats. These threats at the edge

of a reserve include activities around unauthorized trail usage, a high density of invasive species, and a higher susceptibility to human-caused fires [11]. Roads and trails can spread invasive species into rare species habitat and have negatively impacted the recruitment and persistence of rare plants [12,13]. Unfortunately, there are some reserves with high species richness near high-density human populations [14]. These areas are also more vulnerable to increased fire return intervals [15], which negatively impacts some rare species which are sensitive to increased fire regimes [5,6].

Considering the threats and challenges of managing rare plants, it is important that land managers curate and reflect on existing data to make management decisions [16]. Scientists and botanical groups motivated to conserve rare species have curated lists of species that help to rank species based on the type of rarity and the threats to those species within specific geographic areas. Species have been ranked at many spatial scales: globally, through the International Union for Conservation of Nature and Natural Resources [17]; regionally, like the California Native Plant Societies Inventory of Rare Plants [18]; and locally, like a curated list developed by scientists for locally rare species in Napa County [19]. These lists can then be used to develop databases with the occurrence of rare species to inform conservation planning [20].

Incident data within databases can be used to further understand species characteristics or predict the potential presence of new populations [21–23]. However, sampling bias in spatial data can negatively impact rare plant conservation by failing to accurately predict rare species distributions [24]. Spatial sampling biases along roads negatively impact distribution models and reserve planning [25,26]. To account for sampling bias, sampling for rare species should be methodical, covering suitable and unsuitable habitat to create reliable presence-absence data [27], and adequately capture annual variability [28]. Instead, sampling for rare plants is opportunistic considering funding and disturbances like fire or a good rain year following many years of drought. These disturbances only capture part of the environmental variability and leave us questioning whether these rare species are present only after disturbance. While it would be nice to sample more frequently, it is not always feasible for reserve managers to systematically sample an entire reserve annually.

Here, we examine the occurrence and distribution of rare plant species within a natural reserve system in the western United States of America, specifically the Natural Communities Coalition (NCC) Reserve System in Orange County, California. The NCC Reserve System was established under California's Natural Communities Conservation Plan/Habitat Conservation Plan (NCCP/HCP) to broadly protect sensitive species and their habitats adjacent to urban areas in a Mediterranean ecosystem. Within the NCC Reserve System, there are multiple stakeholders and land managers all collecting rare plant data on their individual management units. We seek to understand the distribution of rare plants within the NCC Reserve System, the survey efforts for rare species, their locations in relation to potential threats to their populations, and the vegetation communities these species occur in. Potential threats to rare plant species populations within our study area include disturbances associated with the urban–wildland interface (e.g., increased fire return interval, habitat edge, and habitat patchiness) and changes to habitat type from sea level rise. We expected that rare plant observations would focus on more charismatic species and be near roads and trails due to observer biases and access restrictions for surveyors. We also expected that rare species would be present within all vegetation communities. Still, some vegetation communities such as coastal sage scrub could have higher occurrence records of rare species due to the relative percent coverage of this community type within the protected area. Identifying other vegetation communities that are home to rare plant species will help clarify vegetation types that should be priorities for conservation and restoration efforts. This research aims to further the considerations for rare plant species prioritization and monitoring efforts.

## 2. Materials and Methods

### 2.1. Location Description

The rare plant species within this study occur in Orange County, California, USA and many are protected within the 38,000-acre (15,384 hectares). The NCC Reserve System is dominated by coastal sage scrub habitat and is composed of a Coastal Reserve and a Central Reserve, which are separated by urbanized areas. The land is owned and managed by many different organizations, including County, State, and non-profit entities (Figure 1).

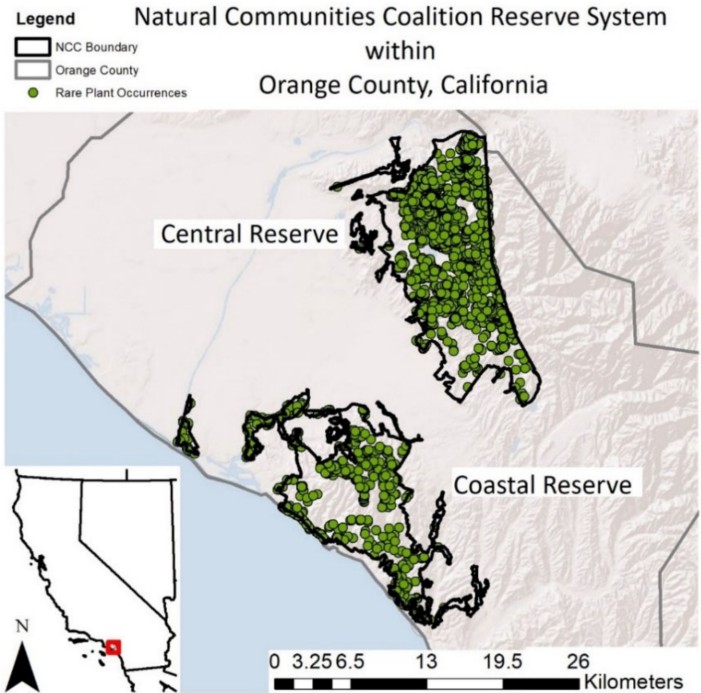

**Figure 1.** Rare plants occur throughout most of the Natural Communities Coalition Reserve System located in Orange County, California. Each point on this map indicates a unique rare plant occurrence record (*N* = 4490).

### 2.2. Species Selection

Ninety-two (92) rare species were selected based on the California Native Plant Society (CNPS) Inventory of Rare Plants database search for Orange County, California and species considered locally rare within Orange County, California. The CNPS Inventory of Rare Plants provides species with a California Rare Plant Rank (CRPR) from one to four, with A or B providing a designation for likely extirpated in California (A), or not extirpated within California (B; CNPS 2019). The last number, 1 through 3, is the threatened status, one indicating the highest threat level. For example, *Brodiaea filifolia* (thread-leaved brodiaea), CRPR 1B.1, is rare, threatened, and endangered in California, currently has an established extent, and is seriously threatened in California with more than 80% of occurrences at a high threat to extirpation. 1B species are already listed or pending state and federal listing under the California Endangered Species Act (CESA) or the Federal Endangered Species Act (ESA). Species with a CRPR of 2 are rare, threatened, or endangered in California but are more common elsewhere, like *Pseudognaphalium leucocephyllum* (white cudweed), which also has occurrences in New Mexico and Arizona. CRPR 3 species are species that CNPS is unsure of sorting into a group due to lack of information about their distribution, threats, ecology, and taxonomy. Lastly, CRPR 4 species are infrequent species with limited distribution, whose populations should be monitored regularly.

This study includes occurrence data from open source websites, password-protected databases, researchers, and consultants. Data for all 92 species were organized into a single geodatabase using ArcGIS 10.5 software by Esri [29]. Data without geographic coordinate information was excluded, as well as all data from before 1900, because those observations were either extirpated or confirmed in a more recent survey. Data were projected using the NAD 1983 UTM Zone 11 coordinate system. The occurrence data was then clipped based on the NCC Reserve System boundary, and only 64 of the 92 species had occurrences within the NCC Reserve System. One of the 65 species is extirpated from the area and was not included in the analyses. Occurrence data were quality controlled by local botanical experts (see Acknowledgements).

## 2.3. Data Analysis

To answer our first question of spatial occurrence for rare plants within the NCC Reserve, we used ArcGIS to map rare plant species occurrence, rare plant species richness, and distance of rare plants from the coastal edge, the urban edge, and the trail edge.

A kernel density tool was used to understand the distribution and density of occurrences throughout the NCC Reserve. Kernel density estimation (KDE) is a non-parametric statistical method for estimating the probability of points per unit area—we chose 100 square meters as the unit area. The KDE calculates a normal distribution for each data point first, and then all the y-values along this distribution are added up. Bumps are created from the added y-values, and the size of the bumps indicates the density of points in that area (larger bumps = higher density) along with the distribution. Probable density distribution is then visualized using a 250-bin color ramp.

To understand the pattern of rare plant species richness across the NCC, we created a grid with an area of 1 square kilometer over the NCC Reserve and joined it to the rare plant occurrence layer by area to obtain a count of the total number of species for each grid polygon. Using the delete identical tool, we removed the species with multiple occurrences per grid cell to bring the value to the number of unique, rare plants per square kilometer. We completed the same steps as richness to obtain samples per square kilometer and occurrence density per square kilometer. To test if rare plant species richness was associated with either the number of samples or the number of occurrences, we used Spearman's rank correlation. We then performed two linear regression analyses to understand if richness could be predicted by the number of samples or number of occurrences.

We used Chi-Square goodness-of-fit and Bonferroni testing to analyze the likelihood of rare plant occurrences within a certain distance from a trail [30], urban edge [31], and or coastal mean high tide line [32]. To answer whether a species was more likely to be found within a specified distance to an edge than not, we created a 50 square meter grid over the NCC Reserve System polygon. We used the total number of squares as the expected value and the number of squares with a rare plant occurrence as the observed value. The distance in meters from an edge (trail, coastline, urban edge) was merged to each 50 square meter polygon. The distance data for each type of edge was then sorted into 11 distance categories. To see the significance between each group and to adjust for Type I error, we calculated the adjusted residuals and compared those values to a new critical level, the z criteria. The z criteria are the inverse of the standard normal cumulative distribution function. If the adjusted residual is greater than or less than the positive and negative z criteria, respectively, we can say that the observed or expected was more or less likely than normal. The Bonferroni test was then verified using a chi-square test to compare one distance category to the compilation of all categories.

We analyzed rare plant species richness by vegetation community for the NCC Reserve System by first intersecting the occurrence data with the vegetation data for the NCC Reserve [31]. To show the density of occurrences, we generated a table that included the count of species occurrences by vegetation community, which was used to obtain species richness. To analyze the association between the acres of each vegetation community and species richness within each vegetation community, we summed the count of species by each vegetation community and performed a Spearman's rank correlation analysis. We ran this test for the entire reserve, and separately for both the coastal the

central reserve. We also used Chi-Square goodness-of-fit and Bonferroni testing to calculate whether the probability that rare species were found in any vegetation type was greater than if the species were randomly distributed across the reserve. We used the polygon count of each vegetation community within the vegetation map to generate the expected values.

To analyze the geographic distribution of the 92 species, we compiled a table with the number of occurrences and the descriptive distribution of the species ranges (Appendix A). We sorted the occurrence data by the total number of occurrences throughout the species' entire range, within Orange County, and within the NCC Reserve. The kernel density maps were used to describe the species range and density of occurrences by area. Rare species ranges were described by the extent of a geographic region like the San Francisco Bay Area, Point Conception, and Baja California. Species ranges were also described with regards to the position of Orange County within the broader range.

## 3. Results

### 3.1. Rare Plant Species Occurrence and Richness

Out of the 92 rare plants in Orange County, 65 of those species occur within the NCC Reserve System (Appendix B, Table A5). *Helianthus nuttallii* ssp. *parishii* (Parish's sunflower), is included in the 65 species, but because it is extirpated from Orange County and likely extinct within its range, it was not included in other analyses. Of the 64 species, there are 4490 rare plant occurrences within the NCC. The distribution of rare plant occurrences is uneven between the Central and Coastal Reserve, with the highest density of rare plant occurrences within the Central Reserve (Figure 2a). Using the occurrence data, we calculated rare plant species richness to range from 0–15 unique species per square kilometer grid throughout the NCC Reserve System (Figure 2b). Rare plant species richness per square kilometer did not differ between the Central and Coastal subregions ($r = 0.90$, $p < 0.05$). However, the Coastal Reserve has the only square kilometer with a rare plant species richness of fifteen. The Coastal Reserve also has a higher rare plant species richness of 53 compared to 39 in the Central Reserve. Twenty-two (22) rare plant species occur in both Reserves, 16 only occur in the Central Reserve and 24 only occur in the Coastal Reserve.

Rare plant species richness per square kilometer grid, number of occurrences per square kilometer grid, and number of samples per one square kilometer were all positively correlated (Table 1). The number of occurrences could significantly predict rare plant species richness, but occurrences only explained 29% of the variation in rare plant species richness by area ($R^2 = 0.29$, $p < 0.001$; Figure 3a). Sampling effort explained more of the variation in species richness by area, but only by 37% ($R^2 = 0.37$, $p < 0.001$; Figure 3b).

**Table 1.** Spearman rank correlation analysis of the rare plant species richness, number of rare plant occurrences, and number of samples all within 1 square kilometer across the Natural Communities Coalition (NCC) Reserve System. All variables were significantly positively correlated. This indicates that species richness increases with sampling effort.

| Correlation | Rho | *p*-Value |
|---|---|---|
| Rare plant species richness per 1 km² by Number of rare plant occurrences per 1 km² | 0.79 | <0.001 |
| Rare plant species richness per 1 km² by Number of samples per 1 km² | 0.71 | <0.001 |
| Number of samples per 1 km² by Number of rare plants occurrences per 1 km² | 0.77 | <0.001 |

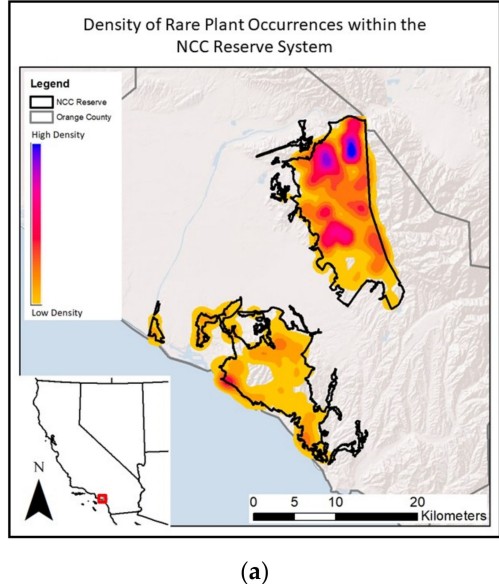

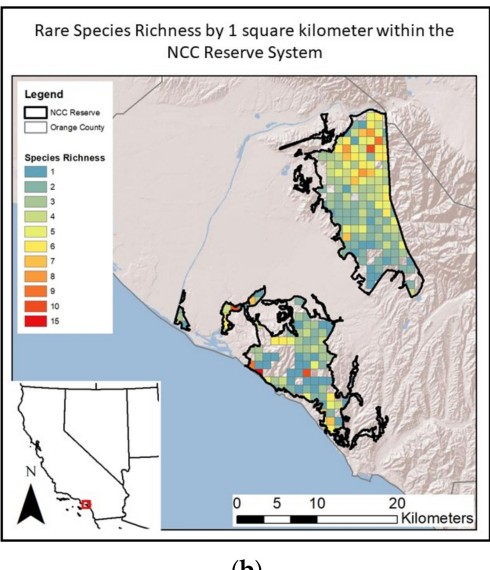

(**a**)                                          (**b**)

**Figure 2.** (**a**) The density of rare plant occurrences within the Natural Communities Coalition Reserve System. Blue indicates a higher density of rare plant occurrences, orange, and yellow indicates a lower density of occurrences. The NE corner of the Central reserve, in the top right corner of the map, has the highest density of occurrences within the Central reserve. (**b**) Rare plant species richness per 1 square kilometer within the Natural Communities Coalition (NCC) Reserve System. The warmer the color, the higher rare plant species richness. White squares indicate that there are no rare plant occurrences within that square km of the NCC black boundary. The coastal reserve contained areas that reached the maximum of 15 rare plants within 1 square kilometer.

Edge Analysis

For all analyses, the likelihood of encountering a rare plant within a specified distance from the urban edge, the coastline, and the edge of a trail was significantly greater than if plants were randomly spread across the NCC Reserve (Appendix B, Table A6).

For the entire NCC Reserve, rare plants were less likely to occur 0–100 m from an urban edge and were more likely to occur 500 m to 5000 m from an urban edge. When analyzed separately, occurrence data within the Central Reserve followed the same pattern of decreased likelihood close to urban edges ($\chi^2$ = 460.28, $p$ < 0.001). This pattern was the opposite within the Coastal Reserve, where the likelihood of encountering a rare plant increased even closer to the urban edge (0–50 m; $\chi^2$ = 53.11, $p$ < 0.001).

The likelihood of finding a rare plant occurrence within a certain distance from the edge of a trail was significantly different than expected for the entire NCC Reserve ($\chi^2$ = 316, $p$ < 0.001). Rare plants were more likely to be observed 0–50 m from the edge of a trail than expected, and less likely to occur farther than 50 m from the edge of a trail. This pattern was similar in the Central reserve, except the likelihood of finding a rare plant more than expected increased up to 400 m. However, after 400 m, rare plant occurrences were less likely to be observed than expected. Like the Central Reserve, rare plants were more likely to be observed within 0–50 m from the edge of a trail than expected ($\chi^2$ = 319.48, $p$ < 0.001). There was no significant relationship between Coastal Reserve rare plants and distances of more than one kilometer from a trail because less than 8% of the Coastal Reserve is more than one kilometer from a trail. Of the total Coastal Reserve, 23% of the area is 0 to 50 m from a trail and 36 of the 53 rare species can be found along the edge of a trail.

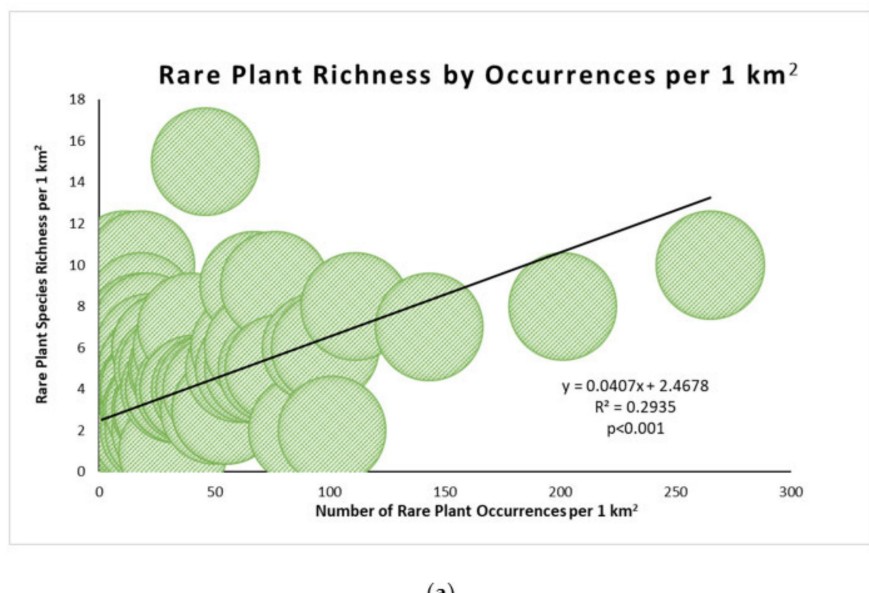

(a)

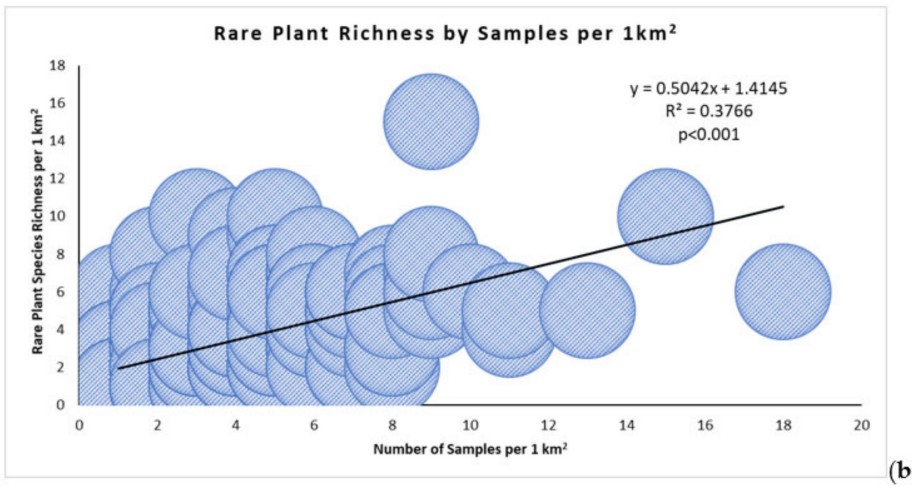

(b)

**Figure 3.** (**a**) Linear regression results of rare plant species richness by rare plant occurrences per one square kilometer in the NCC Reserve System. Each point represents the value of rare plant occurrences and rare plant species richness within a one square kilometer fishnet ($N = 274$). Although significant, occurrences only explained twenty-nine percent of the variation in richness by area. This suggests that other factors, such as environmental heterogeneity, biotic factors, or soil characteristics, may be driving some areas to be diverse hotspots of rarity. (**b**) Linear regression results of rare plant species richness by number of samples per one square kilometer in the NCC Reserve System ($N = 274$). Sampling effort explained 37% of the variation in richness by area.

Coastal Reserve rare plants were more likely to occur within 0 to 200 m from the coastal edge and less likely to occur two to 5 km from the coastline ($\chi^2 = 319.48$, $p < 0.001$). For the entire NCC Reserve, rare plants were more likely to be found either within 500 m of the coast or at least 20 km from the coast, but less likely to be found in the middle ($\chi^2 = 7105$, $p < 0.001$; Table 2).

**Table 2.** Results of chi-square analyses for rare plants per area meters from the coastal edge [1]. Bonferroni post hoc significance values are listed for each category. *p* values with one asterisk (*) indicate that the number of rare species observations was significantly higher than expected (assuming random distribution across the reserve) and *p* values with two asterisks (**) indicate that the number of rare plant occurrences was significantly lower than expected. NS indicates insignificant results.

| Distance from Coastal Edge in Kilometers | Bonferroni Post Hoc Significance Value |
|:---:|:---:|
| 0–0.5 | *p* < 0.004 * |
| 0.5–1 | NS |
| 1–2 | NS |
| 2–3 | *p* < 0.004 ** |
| 3–4 | *p* < 0.004 ** |
| 4–5 | *p* < 0.004 ** |
| 5–10 | *p* < 0.004 ** |
| 10–15 | *p* < 0.004 ** |
| 15–20 | NS |
| 20–30 | *p* < 0.004 ** |

[1] Results of the chi-square analyses for rare plants per area meters from the coastal edge (n = 4491, Degrees of freedom = 10, $\chi^2$ = 7105.33, *p* value = < 0.0001).

### 3.2. Rare Plant Species within Vegetation Communities

Coastal sage scrub is the dominant vegetation community, covering over 50% of the NCC Reserve System, followed by chaparral, and then non-native annual grassland (Figure 4). We found that of all the rare plant occurrences, 40% were within coastal sage scrub. Rare plant occurrences within each vegetation community and the total acreage of each vegetation community were positively correlated (r = 0.89, *p* < 0.01; Table 3). Within the Central Reserve, coastal sage scrub had the highest rare plant occurrences, followed closely by chaparral. No other vegetation communities had rare plant occurrences over 300 (Table 4). Rare plant occurrences within each Central Reserve vegetation community was positively correlated with the total number of acres (r = 0.95, *p* < 0.01). This was also a similar trend in the Coastal Reserve, even with 1490 rare plant occurrences (r = 0.94, *p* < 0.01). Forty-eight (48) percent of the Coastal Reserve rare plant occurrences are within coastal sage scrub, which makes up 50% of the total vegetation community within the Coastal reserve.

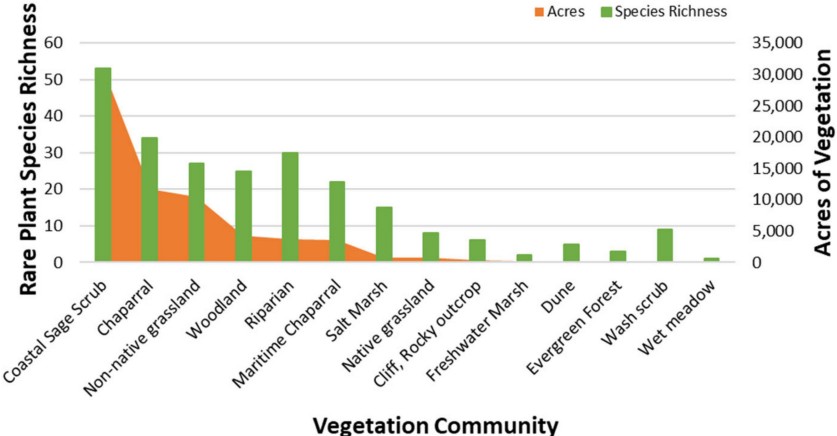

**Figure 4.** Rare plant species richness within each vegetation community with acres of each vegetation type within the Natural Communities Coalition Reserve System. Coastal sage scrub has the highest acreage and the highest rare plant species richness of 53.

**Table 3.** Spearman rank correlation analysis of the acreage of vegetation community by the occurrences of rare plants within each vegetation community and the rare plant species richness within each vegetation community. All analyses were significantly positively correlated. This indicates that occurrences and richness increase with the increasing size of each vegetation community.

| Correlation | Rho | *p*-Value |
|---|---|---|
| Total rare plant occurrences per Vegetation Community, Acres of Vegetation Community | 0.89 | <0.01 |
| Total rare plant species richness per Vegetation Community, Total Acres of Vegetation Community | 0.90 | <0.01 |
| Coastal rare plant occurrences per Vegetation Community, Coastal Acres of Vegetation Community | 0.94 | <0.01 |
| Coastal rare plant species richness per Vegetation Community, Coastal Acres of Vegetation Community | 0.95 | <0.01 |
| Central rare plant occurrences per Vegetation Community, Central Acres of Vegetation Community | 0.95 | <0.01 |
| Central rare plant species richness per Vegetation Community, Central Acres of Vegetation Community | 0.98 | <0.01 |

**Table 4.** Rare plant occurrences and rare plant species richness by acres of each vegetation community within the Natural Communities Coalition Reserve System.

| Vegetation Community | Acres | Rare Plant Occurrences | Species Richness |
|---|---|---|---|
| Coastal sage scrub | 30,825 | 1783 | 53 |
| Chaparral | 11,719 | 1445 | 34 |
| Non-native grassland | 10,429 | 365 | 27 |
| Woodland | 4272 | 116 | 25 |
| Riparian | 3680 | 147 | 30 |
| Maritime chaparral | 3584 | 255 | 22 |
| Saltmarsh | 781 | 49 | 16 |
| Native grassland | 745 | 158 | 8 |
| Cliff, rocky outcrop | 350 | 25 | 6 |
| Freshwater marsh | 182 | 4 | 2 |
| Dune | 122 | 12 | 5 |
| Evergreen forest | 39 | 5 | 3 |
| Wash scrub | 31 | 36 | 9 |
| Wet meadow | 30 | 1 | 1 |

We found that coastal sage scrub has the highest rare plant species richness with 53 unique, rare plants out of the total 34 rare species within the NCC Reserve (Figure 5). The second-highest rare plant species-rich community was in chaparral followed by riparian for all 14 vegetation communities. Non-native annual grassland had the fourth-highest species richness with 27 rare species. NCC rare plant species richness was positively correlated by the acreage of each vegetation community (r = 0.90, *p* < 0.01). We also found this to be similar when separating rare plant species richness within each vegetation community by the Coastal (r = 0.95, *p* < 0.01), and Central Reserve (r = 0.98, *p* < 0.01). The coastal sage scrub vegetation community had higher rare plant species richness in the Coastal Reserve compared to that of Central Reserve, 36 to 30, respectively (Figure 5; Appendix B, Table A7). For the Coastal and Central Reserve, non-native annual grassland had a species richness of 19—the second most rare species-rich vegetation community for the Coastal Reserve. Saltmarsh vegetation community only occurs within the Coastal Reserve and has a rare plant species richness of 14. Rare plant species richness for woodlands was higher in the Central Reserve than in the Coastal Reserve, but the Central Reserve has 2000 more acres of woodland.

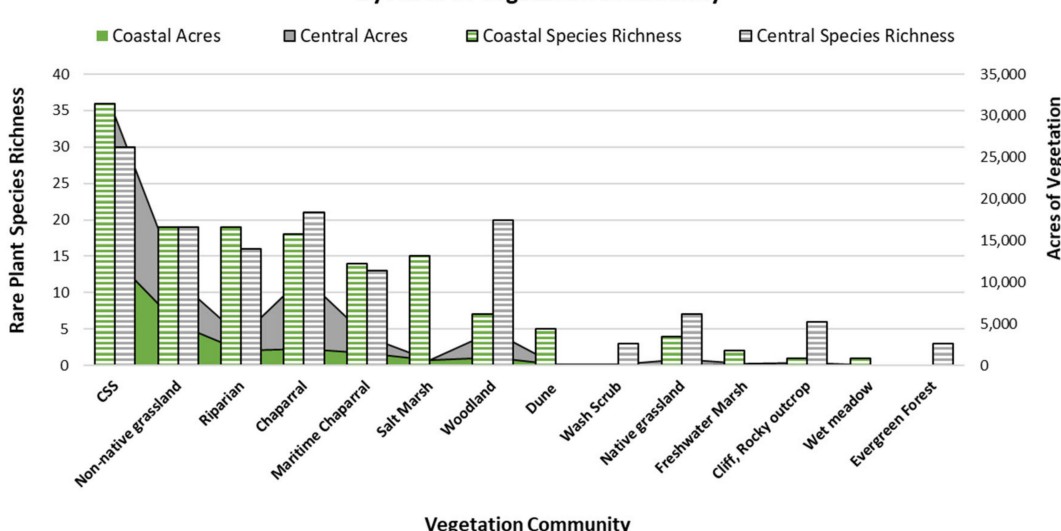

**Figure 5.** Central and Coastal reserve rare plant species richness by acres of vegetation community within each reserve subregion.

The likelihood of finding a rare plant within all vegetation communities was significantly different than expected based on acreage alone ($\chi^2 = 2057$, $p < 0.001$). However, Bonferroni testing showed that the direction of the results (more or less rare plants than expected) were different between the vegetation communities, and not all results were significant (Appendix B, Table A8). Based on the count of each vegetation community polygon, rare plants were less likely to be observed in evergreen forests, freshwater marshes, wet meadows, and on rocky outcrops.

3.2.1. Rare Plant Species Distributions

The rare plants within the NCC Reserve system have diverse ranges. Thirteen species have their northern range within the NCC Reserve and Orange County (Appendix A, Table A1). Two of those species, *Lepechinia cardiophylla* (Santa Ana pitchersage) and *Microseris douglasii* ssp. *platycarpha* (Douglas' silverpuffs), have a high density of occurrences within Orange County (Appendix B, Table A5). Orange County captures almost 50% of *Lepechinia cardiophylla* occurrences, with 13% of the total occurrences within the NCC Reserve. *Microseris douglasii* ssp. *platycarpha* has less than 50% of its total occurrences in Orange County, with 16% of the total occurrences within the NCC (Appendix C, Figure A1). Both species have ranges that extend from Orange County to Northern Baja.

There are 19 rare species that have their Southern ranges within the NCC Reserve and Orange County. These species vary in the distribution of their ranges, from North and South of Point Conception to Orange County or North to the Oregon border (Appendix A, Table A2). Five of those species have their highest density of occurrences within Orange County. These species include *Atriplex davidsonii* (Davidson's saltscale), *Calandrinia breweri* (Brewer's red maids), *Calochortus catalinae* (Catalina Mariposa Lily), *Dudleya cymosa* ssp. *ovatifolia* (Santa Monica mountains dudleya) and *Phacelia hubbyi* (Hubby's phacelia). The NCC Reserve captures more than half of the total recorded occurrences for *Calochortus catalinae* and *Dudleya cymosa* ssp. *ovatifolia* (Appendix C, Figure A1).

A majority of Orange County's rare plant species have the center of their range within Orange County and the NCC Reserve. Of the 29 species, 17 have their highest density of occurrences within Orange County (Appendix A, Table A3). Orange County has one endemic species and one endemic subspecies, *Dudleya stolonifera* (Laguna Beach dudleya) and *Pentachaeta aurea* ssp. *allenii* (Allen's daisy), respectively. Unlike *Dudleya stolonifera*, which has all its occurrences within the NCC Reserve, *Pentachaeta aurea* ssp. *allenii* has occurrences outside of the NCC reserve. A majority of all the

occurrences for *Calochortus weedii var. intermedius* (intermediate mariposa lily), *Dudleya multicaulis* (many-stemmed dudleya), *Nolina cismontane* (peninsular beargrass), and *Verbesina dissita* (crownbeard) are within the NCC Reserve. *Verbesina dissita* is found in Orange County and Baja, skipping San Diego County, making the NCC Reserve System a significant area for the conservation of this species.

With restricted ranges from north to south, there are three species within Orange County that are more longitudinally distributed, with their western extant reaching Orange County (Appendix A, Table A4). *Hordeum intercedens* (Vernal barley) occurs from Orange County to Riverside, while *Lycium californicum* (California boxthorn) has a broader east to west range from South Point Conception to New Mexico. *Penstemon californicus* (California beardtongue) has the northwestern boundary of its range reaching into Orange County. *Hordeum intercedens* and *Lycium californicum* have their highest density of occurrences within Orange County, and most of those occurrences are in the NCC reserve. However, many *Lycium californicum* occurrences are not within Orange County but spread throughout its range.

### 3.2.2. Species Occurrences and Rarity

The number of occurrences per rare plant species rank was extremely variable (Figure 6). *Dudleya cymosa* ssp. *ovatifolia* had the minimum number of rare plant occurrences, with ten occurrences throughout its range (Appendix C, Figure A2). On the other hand, *Quercus dumosa* (scrub oak) had 1127 plant occurrences, the most of any of the rare species. Within the NCC Reserve, the lowest number of occurrences was one occurrence for *Atriplex parishii* (Parish's brittlescale), *Lilium humboldtii* ssp. *ocellatum* (Humboldt lily), *Navarretia prostratea* (prostrate navarretia), *Penstemon californicus*, and *Symphyotrichum defoliatum* (San Bernardino Aster). *Dudleya multicaulis* had the most occurrence points with 710 occurrences throughout the NCC Reserve. The average number of rare plant occurrences per species within the NCC Reserve was 39, but most species have less than 39 occurrences. However, the number of rare plant species occurrences per rarity rank compared to the number of rare species within each rarity rank was positively correlated, indicating that the type of rarity did not contribute to oversampling ($r = 0.89$, $p < 0.05$; Figure 6).

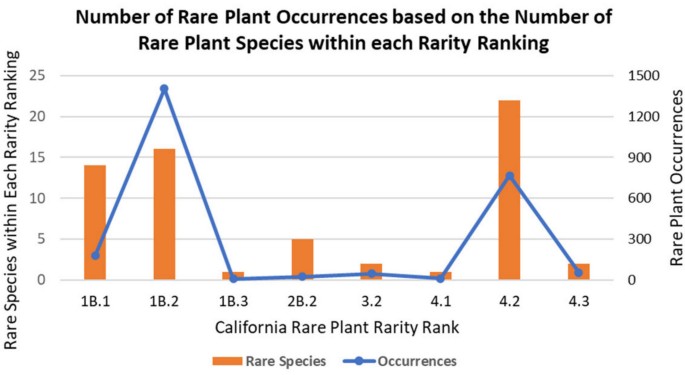

**Figure 6.** Number of rare plant occurrences within each CRPR ranking, compared to the total number of rare plant species that fall into the CRPR category. Out of the 64 rare species within the NCC reserve, 30 rare species are in rank one with 14 species being extremely threatened. Twenty-two species are in rank 4.

## 4. Discussion

### 4.1. Rare Plant Species Occurrence and Richness

Management and monitoring of rare species within protected areas requires comprehensive and continuous sampling to reach current biodiversity goals and inform future management decisions [16].

To assist management priorities, we created a collective database of rare plant occurrence data to analyze rare plant species distributions and spatial patterns with other variables of reserve management concern. Sampling effort did not strongly predict rare plant species richness and explained only some of the variation. This may be due to the type of sampling data within the database which was a mixture of targeted sampling efforts from contracted botanists and community science data from opportunistic botanists. A targeted survey effort would only indicate one sample over a larger area, while the occurrences recorded from opportunistic botanists could be frequent or infrequent across a landscape, depending on access to the land. However, repeated surveys from contractors or land managers may not be feasible based on time and financial capacity. Land managers can supplement contractor surveys with opportunistic sampling to collect more information on species presence, which can influence species richness within an area. On the other hand, contractor surveys can be used to fill key data gaps created within opportunistic sampling efforts from individual community botanists [33]. Similar studies estimating species richness using databases have shown that even one extensive in-situ survey could not accurately predict species richness as well as database records did for the area [34]. Additionally, opportunistic observations paired with focused sampling effort can increase the capacity to estimate species richness [35]. Our results continue to support that paired contractor and community science observations can influence species detection efforts. Rare plant species richness could also be explained from the extent of the sampling grid. Increasing grain size increases the possibility of more area captured and thus increases species richness [36,37]. However, regardless of size, some areas could have high rare species richness because they are an ecotone of unique habitats [38].

Edge Analysis

Our mapping efforts show that many rare species were more likely to be observed near a coastal, urban, or a road edge. This is troubling for rare coastal plants because these species are particularly vulnerable to sea-level rise due to climate change [39]. Sea level rise (SLR) is currently increasing by 3.3 mm per year and is projected to rise 65 cm by 2100 [40]. Even under low SLR scenarios, high marsh habitats will be submerged by the end of the century [41]. Habitat that currently supports *Chloropyron maritimum* ssp. *maritimum* (salt marsh bird's beak) and *Juncus acutus* ssp. *leopoldii* (wiregrass) is projected to become mudflats and a coastal occurrence of *Phacelia hubbyi* could become extirpated due to rising waters. Species along the coastal bluffs that occur within 0–50 m from mean high tide lines are also threatened from increased rates of erosion. These species include *Atriplex davidsonii*, *Dudleya multicaulis*, *Centromadia parryi* ssp. *australis* (southern tarplant), *Bahiopsis laciniata* (San Diego viguiera), *Aphanisma blitoides* (Aphanisma), *Atriplex pacifica* (south coast saltscale), *Atriplex coulteri* (Coulter's saltbush), *Lycium californicum*, *Suaeda esteroa* (Estuary seablite ), and *Suaeda taxifolia* (seablite). Furthermore *Atriplex pacifica* only occurs on the bluff within 0–50 m of the mean high tide line, and *Abronia maritima* (sticky sand verbena) only occurs in sandy areas 0–50 m from the mean high tide line. Based on models of impacts from sea level rise, many coastal rare plants are threatened by inundation, flooding, and dune and cliff erosion [39,42].

Rare plants were less likely to occur near the urban edge within the Central Reserve, which could be due to the impacts of edge effects [11,43]. The increase in anthropogenic influenced wildfires is an increasing threat at the urban edge within our study area. The Central Reserve has had four significant fires which have burned at least half of the entire Central Reserve in the last 40 years, while the Coastal Reserve has had one significant and a few small fires in the last 40 years [44]. Wildfires present opportunities to survey for disturbance dependent species and species that require fire for germination [45]. However, increases in the fire return interval due to proximity to high urban population densities also have negative impacts to rare species and their vegetation communities [46,47]. Modeling for an increased fire return interval of more than 40 years has shown to negatively affect one of the rare species within our database, *Hesperocyparis forbesii* (Tecate Cypress ) [48], and there have already been 4 fires within that time span. While some rare species may be able to exist within small patchy habitats [8,49] the disturbance regime and the species tolerance for disturbance within the

surrounding area will influence the long-term success of those rare species [50]. Monitoring rare plant species near the urban edge could help explain the impacts to the habitat and how species respond to an increased disturbance regime.

The patterns of rare plant species richness, occurrence, and sampling in relation to other variables support our first hypothesis that rare plants within the NCC Reserve are more likely to occur along roads and trails. Although the current sampling was able to capture high rare species richness within the NCC Reserve System, some areas have never been sampled. Within the Coastal Reserve, road and trail sampling was apparent, however, the Coastal Reserve has less area overall and even less area away from a trail. On the other hand, the Central Reserve is much larger, but had more rare plant occurrences sampled near a road or trail than the farthest distances from a road or trail. Road sampling bias is not uncommon within the botanical sampling literature [51]. Sampling bias along roads has serious implications on the type of rare species captured as roads and trails have higher rates of disturbance. Rare species sensitive to disturbance and further away from roads could be underrepresented within the database. These species could also just be unfortunate to occur next to a road and should be prioritized for monitoring to evaluate the impacts of high disturbance [52].

The habitat for rare species within our study site is fragmented not only by urban areas but also by roads and trails within the protected habitat. Nature reserves established to protect the threatened species within are finding that fragmentation is leading to decreases in rare plant populations [53], which highlights a further need to closely monitor rare plant populations within nature reserves. Habitat fragmentation caused by physical barriers like an urban edge or a coastline create closed systems that prevent species from moving along geographical gradients, which can drive species to local extinction [54]. Land managers may be challenged to manage habitat designated for rare plant species that may no longer be suitable for the species under a changing climate. Conservation strategies could shift to ex-situ conservation for future out-planting [55–57] and even assisted migration throughout patches of protected habitat in other latitudes [58]. However, this may not be successful for all species based on their habitat specificity [59].

### 4.2. Rare Plant Species within Vegetation Communities

Our second research question asked what vegetation communities rare plants occur in and which communities support the greatest number of rare plant species. Our findings confirm our predictions that the coastal sage scrub vegetation community hosts 82% of the rare species within the NCC Reserve because that habitat type is the most common, taking up the majority of the area of the Reserve. Coastal sage scrub also provides habitat for many other sensitive species [60]. This diverse habitat type is threatened by an increased disturbance regime and proximity to urban development [46]. Manipulations of water and nitrogen following wildfire within the NCC Reserve demonstrated that drought and nitrogen deposition post-fire caused coastal sage scrub to convert to non-native dominated annual grassland [61]. However, vegetation monitoring within the Reserve indicated that coastal sage scrub was resilient to one recent wildfire when followed by years with high rainfall and to a separate extended drought event, and can be resistant to vegetation type conversion [62].

The most surprising result was the high number of rare species within non-native vegetation communities. These communities can alternate annually between high non-native grass cover and invasive forbs like *Brassica nigra* (black mustard) [63]. Non-native annual grasslands also host many native forbs, some of which only occur in wet years or post-fire [62]. Overall, invasive plant species negatively impact rare plant species populations [13,64,65], and therefore rare plant species benefit from the removal of non-native and invasive species [60,66,67]. Restoration of non-native vegetation communities to either native coastal sage scrub or native grasslands should include pre-assessment of the rare species on-site to prevent extirpation from restoration related impacts.

Rare Plant Species Distributions

Orange County and the NCC Reserve System host species with diverse ranges and occurrences. The Reserve System captures species on the periphery of their ranges and species whose ranges are centered within the study area, highlighting the importance of the placement of the Reserve. Although we did not find any correlation between the number of rare species per rare plant rank and the count of occurrences per rare plant rank, three species make up 60% of all occurrences within the database. *Dudleya multicaulis* and *Calochortus weedii var. intermedius* are both the rarest ranked species, with small ranges and the center of their ranges are within Orange County. On the other hand, *Calochortus catalinae* is not as rare compared to *Dudleya multicaulis* and *Calochortus weedii var. intermedius* but has a lot more occurrences than any other species within the same rank category. With occurrence data, it is unclear if each incidence represents a single plant or an entire population; however, occurrence data cannot be used as abundance data. These species are abundant throughout their range, both *Calochortus* species are common after a fire, but all three are also particularly charismatic. Charismatic species are often oversampled, which can negatively impact species that may be rarer that are less charasmatic [68]. Sampling bias has also been found to negatively impact the knowledge of extinct flora [69]. Sampling towards more charismatic species is not uncommon outside of plant conservation [33,70] and has also been found to influence teaching of conversation biology [71].

## 5. Conclusions

Comprehensive analysis of the rare plant occurrences within our study region revealed geographic locations (e.g., coastal bluffs and farther away from trails) and vegetation types (e.g., invaded grasslands, coastal sage scrub, and salt marsh) with high numbers of rare plants, as well as, species that are endemic (e.g., *Dudleya stolonifera* and *Pentachaeta aurea* ssp. *allenii*) or whose ranges fall mostly within the Reserve (e.g., *Dudleya multicaulis* and *Calochortus weedii var. intermedius*). The data also revealed undocumented bias by species and uneven sampling throughout our study region. Although the species within our project area are protected, they are still vulnerable to threats, such as sea-level rise, increased disturbance regime, and habitat fragmentation. Land managers can use rare plant occurrence data to assess species distributions across multiple scales to prioritize species monitoring based on priority threats. However, occurrence data can include sampling bias and focused survey efforts may be necessary to address data gaps. We recommend that rare plant occurrence data be continually updated, and that long-term monitoring plots be established for the rarest species.

**Author Contributions:** Conceptualization, H.L. and S.K.; methodology, H.L.; validation, H.L., S.K. and E.D.C.; formal analysis, H.L.; investigation, H.L.; resources, E.D.C. and S.K.; data curation, H.L.; writing—original draft preparation, H.L.; writing—review and editing, H.L.; visualization, H.L., S.K. and E.D.C.; supervision, S.K. and E.D.C.; project administration, E.D.C. and S.K.; funding acquisition, H.L. All authors have read and agreed to the published version of the manuscript.

**Funding:** This research was funded by the Center for Environmental Biology at the University of California, Irvine (UCI), which receives funding from the Natural Communities Coalition and The Nature Conservancy. Additional support was provided by UCI's Masters in Conservation and Restoration Science program, which receives funding from the Voth Foundation.

**Acknowledgments:** We thank Fred Roberts, Ron Vanderhoff, and Bob Allen for their contributions to data and data quality. We thank the undergraduate students who helped with data organization: Melissa Stellar, Emilie Chien, Yueqi Gu, Bryan Lam, Victor Paitimusa, Jocelyne De La Torre Agustin, and Noel Leanos-Mejia. We would also like to thank the staff at the Natural Communities Coalition, Travis Huxman, and the NCC land managers within the NCC Reserve System for providing data and input. The Nature Reserve of Orange County is located on the ancestral homeland of the Acjachemen and Tongva Peoples.

**Conflicts of Interest:** The design of this study was informed by workshops and conversations with local botanists and with members of the Natural Communities Coalition (NCC), which provided much of the funding for this research.

## Appendix A

**Table A1.** Rare Species that occur within the Natural Communities Coalition (NCC) Reserve System and have their northern ranges extend into Orange County.

| Species | Highest Density of Occurrences | Range |
|---------|-------------------------------|-------|
| *Artemisia palmeri* | San Diego County | Orange County to North Baja |
| *Bahiopsis laciniata* | San Diego County | South Point Conception to Baja |
| *Chorizanthe polygonoides var. longispina* | Riverside County | Orange County to San Diego |
| *Comarostaphylis diversifolia* ssp. *diversifolia* | San Diego County | South Point Conception to North Baja |
| *Euphorbia misera* | San Diego County | Orange County to South Baja |
| *Harpagonella palmeri* | Riverside County | Los Angeles County to South Baja |
| *Iva hayesiana* | San Diego County | Orange County to San Diego |
| *Lepechinia cardiophylla* | Orange County | Orange County to North Baja |
| *Microseris douglasii* ssp. *platycarpha* | Orange County | Orange County to North Baja |
| *Pentachaeta aurea* | Riverside County | San Bernardino County |
| *Selaginella cinerascens* | San Diego County | Orange County to North Baja |
| *Suaeda esteroa* | San Diego County | South Point Conception to Baja |
| *Symphyotrichum defoliatum* | San Diego County | North Point Conception to South San Diego |

**Table A2.** Rare Species that occur within the NCC Reserve System and have their Southern ranges extend into Orange County.

| Species | Highest Density of Occurrences | Range |
|---------|-------------------------------|-------|
| *Astragalus brauntonii* | Ventura County | Ventura County to Orange County |
| *Atriplex davidsonii* | Orange County | North Point Conception to Orange County |
| *Atriplex parishii* | Riverside County | Bay Area to Riverside County |
| *Baccharis malibuensis* | Los Angeles County | Los Angeles County to Orange County |
| *Calandrinia breweri* | Orange County | California |
| *Calochortus catalinae* | Orange County | North Point Conception to Orange County |
| *Calochortus plummerae* | Los Angeles County | Peninsular Ranges |
| *Chorizanthe parryi var. fernandina* | Los Angeles County | Los Angeles to Orange County |
| *Convolvulus simulans* | Fresno County | Bay Area to North Baja |
| *Dudleya cymosa* ssp. *ovatifolia* | Orange County | Los Angeles County to Orange County |
| *Horkelia cuneata var. puberula* | Santa Barbara County | North Point Conception to San Diego County |
| *Juglans californica* | Los Angeles County | California |
| *Malacothrix saxatilis var. saxatilis* | Santa Barbara County | South Point Conception to Orange County |
| *Navarretia prostrata* | Fresno County | Bay Area to San Diego County |
| *Phacelia hubbyi* | Orange County | South Point Conception to Orange |
| *Phacelia ramosissima var. austrolitoralis* | Santa Barbara County | Monterey Bay to North Baja |
| *Quercus dumosa* | San Diego County | California |
| *Quercus engelmannii* | Los Angeles County | Los Angeles County to North Baja |
| *Senecio aphanactis* | Fresno County | California |

**Table A3.** Species with ranges that have the NCC Reserve and Orange County as their center.

| Species | Highest Density of Occurrences | Range |
|---------|-------------------------------|-------|
| *Abronia maritima* | Santa Barbara County | Bay area to north Baja |
| *Aphanisma blitoides* | San Clemente island | North Point Conception to San Diego County |
| *Atriplex coulteri* | Orange County | North Point Conception to San Diego County |
| *Atriplex pacifica* | San Clemente island | South Point Conception to San Diego County |
| *Brodiaea filifolia* | Orange County | Orange County to San Diego County |
| *Calochortus weedii var. intermedius* | Orange County | Santa Ana mountains |
| *Centromadia parryi* ssp. *australis* | Orange County | South Point Conception to norther Baja |
| *Chloropyron maritimum* ssp. *maritimum* | San Luis obispo County | North Point Conception to San Diego County |
| *Deinandra paniculata* | Riverside County | North Point Conception to north Baja |
| *Dichondra occidentalis* | San Diego County | North Point Conception to north Baja |
| *Dudleya multicaulis* | Orange County | Los Angeles County to San Diego |
| *Dudleya stolonifera* | Orange County | Orange County |

**Table A3.** *Cont.*

| Species | Highest Density of Occurrences | Range |
|---|---|---|
| *Hesperocyparis forbesii* | Orange County | Los Angeles County to San Diego County |
| *Juncus acutus* ssp. *leopoldii* | San Diego County | North Point Conception to San Diego County |
| *Lasthenia glabrata* ssp. *coulteri* | Riverside County | North Point Conception to San Diego County |
| *Lepidium virginicum* var. *robinsonii* | Orange County | South Point Conception to San Diego County |
| *Lilium humboldtii* ssp. *cellatum* | Riverside County | South Point Conception to San Diego County |
| *Monardella hypoleuca* ssp. *intermedia* | Orange County | Orange County to San Diego County |
| *Nama stenocarpum* | Orange County | Orange County to south Baja and Texas |
| *Nemacaulis denudata* var. *denudata* | San Diego County | North Point Conception to north Baja |
| *Nolina cismontana* | Orange County | South Point Conception to San Diego County |
| *Ophioglossum californicum* | San Diego County | Central valley to south Baja |
| *Pentachaeta aurea* ssp. *allenii* | Orange County | Orange County |
| *Piperia cooperi* | Catalina island | North Point Conception to north Baja |
| *Polygala cornuta* var. *fishiae* | Riverside, San Diego, and Orange County border | South Point Conception to south Baja |
| *Pseudognaphalium leucocephalum* | San Diego County/Orange County border | Bay area to Texas |
| *Romneya coulteri* | Orange County | South Point Conception to north Baja |
| *Suaeda taxifolia* | Orange County | North Point Conception to north Baja |
| *Verbesina dissita* | Orange County | Orange County to Baja |

**Table A4.** Rare species located in the NCC Reserve that are more longitudinally distributed having a limited North to South range.

| Species | Highest Density of Occurrences | Range |
|---|---|---|
| *Hordeum intercedens* | Orange County | Orange County and Riverside County |
| *Lycium californicum* | Orange County | South Point Conception to New Mexico |
| *Penstemon californicus* | Riverside County | Riverside County to North Baja |

## Appendix B

**Table A5.** Rare plant species ordered alphabetically within the Natural Communities Coalition (NCC) Reserve System and their presence or absence between the two subregions—Coastal and Central—and their California native plant society California rare plant rank (CRPR). Out of the 64 species, 22 species occur in both subregions, 27 species occur only in the coastal reserve, 16 only occur in the central reserve. *Helianthus nuttallii* ssp. *parishii* (Parish's sunflower), which is underlined in the table, is the only species extirpated within the NCC Reserve and is presumed extinct throughout its range.

| Rare Plant Species | Coastal | Central | CRPR |
|---|---|---|---|
| *Abronia maritima* | Yes | No | 4.2 |
| *Aphanisma blitoides* | Yes | No | 1B.2 |
| *Artemisia palmeri* | Yes | No | 4.2 |
| *Astragalus brauntonii* | No | Yes | 1B.1 |
| *Atriplex coulteri* | Yes | No | 1B.2 |
| *Atriplex pacifica* | Yes | No | 1B.2 |
| *Atriplex parishii* | Yes | No | 1B.1 |
| *Atriplex davidsonii* | Yes | No | 1B.2 |
| *Baccharis malibuensis* | No | Yes | 1B.1 |
| *Bahiopsis laciniata* | Yes | Yes | 4.2 |
| *Brodiaea filifolia* | Yes | Yes | 1B.1 |
| *Calandrinia breweri* | No | Yes | 4.2 |
| *Calochortus catalinae* | Yes | Yes | 4.2 |

**Table A5.** *Cont.*

| Rare Plant Species | Coastal | Central | CRPR |
|---|---|---|---|
| *Calochortus plummerae* | No | Yes | 4.2 |
| *Calochortus weedii* var. *intermedius* | Yes | Yes | 1B.2 |
| *Centromadia parryi* ssp. *australis* | Yes | No | 1B.1 |
| *Chloropyron maritimum* ssp. *maritimum* | Yes | No | 1B.2 |
| *Chorizanthe parryi* var. *fernandina* | No | Yes | 1B.1 |
| *Chorizanthe polygonoides* var. *longispina* | No | Yes | 1B.2 |
| *Comarostaphylis diversifolia* ssp. *diversifolia* | Yes | No | 1B.2 |
| *Convolvulus simulans* | Yes | Yes | 4.2 |
| *Deinandra paniculata* | Yes | Yes | 4.2 |
| *Dichondra occidentalis* | Yes | No | 4.2 |
| *Dudleya cymosa* ssp. *ovatifolia* | No | Yes | 1B.2 |
| *Dudleya multicaulis* | Yes | Yes | 1B.2 |
| *Dudleya stolonifera* | Yes | No | 1B.1 |
| *Euphorbia misera* | Yes | No | 2B.2 |
| *Harpagonella palmeri* | Yes | Yes | 4.2 |
| *Helianthus nuttallii* ssp. *parishii* | <u>Yes</u> | <u>No</u> | <u>1A</u> |
| *Hesperocyparis forbesii* | No | Yes | 1B.1 |
| *Hordeum intercedens* | Yes | Yes | 3.2 |
| *Horkelia cuneata* var. *puberula* | Yes | Yes | 1B.1 |
| *Iva hayesiana* | Yes | No | 2B.2 |
| *Juglans californica* | Yes | Yes | 4.2 |
| *Juncus acutus* ssp. *leopoldii* | Yes | No | 4.2 |
| *Lasthenia glabrata* ssp. *coulteri* | Yes | Yes | 1B.1 |
| *Lepechinia cardiophylla* | No | Yes | 1B.2 |
| *Lepidium virginicum* var. *robinsonii* | Yes | Yes | 4.3 |
| *Lilium humboldtii* ssp. *ocellatum* | No | Yes | 4.2 |
| *Lycium californicum* | Yes | No | 4.2 |
| *Malacothrix saxatilis* var. *saxatilis* | Yes | Yes | 4.2 |
| *Microseris douglasii* ssp. *platycarpha* | Yes | Yes | 4.2 |
| *Monardella hypoleuca* ssp. *intermedia* | No | Yes | 1B.3 |
| *Nama stenocarpum* | Yes | No | 2B.2 |
| *Navarretia prostrata* | Yes | No | 1B.1 |
| *Nemacaulis denudata* var. *denudata* | Yes | No | 1B.2 |
| *Nolina cismontane* | No | Yes | 1B.2 |
| *Ophioglossum californicum* | Yes | Yes | 4.2 |
| *Penstemon californicus* | No | Yes | 1B.2 |
| *Pentachaeta aurea* | Yes | Yes | N/A |
| *Pentachaeta aurea* ssp. *allenii* | Yes | Yes | 1B.1 |
| *Phacelia hubbyi* | No | No | 4.2 |
| *Phacelia ramosissima* var. *austrolitoralis* | Yes | No | 3.2 |
| *Piperia cooperi* | No | Yes | 4.2 |
| *Polygala cornuta* var. *fishiae* | Yes | Yes | 4.3 |
| *Pseudognaphalium leucocephalum* | No | Yes | 2B.2 |
| *Quercus dumosa* | Yes | Yes | 1B.1 |
| *Quercus engelmannii* | Yes | Yes | 4.2 |
| *Romneya coulteri* | No | Yes | 4.2 |
| *Selaginella cinerascens* | Yes | No | 4.1 |
| *Senecio aphanactis* | Yes | Yes | 2B.2 |
| *Suaeda esteroa* | Yes | No | 1B.2 |
| *Suaeda taxifolia* | Yes | No | 4.2 |
| *Symphyotrichum defoliatum* | Yes | No | 1B.2 |
| *Verbesina dissita* | Yes | No | 1B.1 |

**Table A6.** Results of multiple chi-square analyses for rare plants per area meters from an edge. Bonferroni post hoc significance values are listed for each category. *p* values with one asterisk (*) indicate that the number of rare species observations were significantly higher than expected (assuming random distribution across the reserve) and *p* values with two asterisks (**) that the number of rare plant occurrences were significantly lower than expected. NS indicates insignificant results; NA means there was not data that fell into that category, so the test was not run.

| | NCC to Urban Edge | Central Reserve to Urban Edge | Coastal Reserve to Urban Edge | NCC: Trail Edge | Central Reserve: Trail Edge | Coastal Reserve: Trail Edge | Coastal Reserve: Coastline |
|---|---|---|---|---|---|---|---|
| *n* | 2915 | 2401 | 514 | 2915 | 2401 | 514 | 514 |
| Degrees of freedom | 10 | 10 | 8 | 11 | 10 | 11 | 11 |
| $\chi^2$ | 723.34 | 460.28 | 53.11 | 316 | 1476 | 110.97 | 1476 |
| *p* value | $p < 0.001$ | $p < 0.001$ | $p < 0.001$ | $p < 0.001$ | $p < 0.001$ | $p < 0.001$ | $p < 0.001$ |
| Distance categories (meters) | **Bonferroni post-hoc significance value** | | | | | | |
| 0–50 | $p < 0.004$ ** | $p < 0.004$ ** | NS | $p < 0.004$ * | $p < 0.004$ * | $p < 0.004$ * | $p < 0.004$ * |
| 50–100 | $p < 0.004$ ** | $p < 0.004$ ** | $p < 0.005$ * | $p < 0.004$ ** | $p < 0.004$ * | $p < 0.004$ ** | $p < 0.004$ * |
| 100–200 | NS | NS | $p < 0.005$ * | $p < 0.004$ ** | $p < 0.004$ * | NS | $p < 0.004$ * |
| 200–300 | NS | NS | $p < 0.005$ * | NS | $p < 0.004$ * | $p < 0.004$ ** | NS |
| 300–400 | NS | NS | NS | NS | $p < 0.004$ * | NS | NS |
| 400–500 | NS | NS | NS | NS | NS | NS | NS |
| 500–1000 | $p < 0.004$ | $p < 0.004$ * | NS | NS | $p < 0.004$ ** | $p < 0.004$ ** | NS |
| 1000–2000 | $p < 0.004$ | $p < 0.004$ * | $p < 0.005$ ** | $p < 0.004$ ** | $p < 0.004$ ** | NS | NS |
| 2000–3000 | $p < 0.004$ | $p < 0.004$ * | NS | $p < 0.004$ ** | $p < 0.004$ ** | NS | $p < 0.004$ ** |
| 3000–4000 | $p < 0.004$ | NS | NA | $p < 0.004$ ** | $p < 0.004$ ** | NS | $p < 0.004$ ** |
| 4000–5000 | $p < 0.004$ | $p < 0.004$ * | NA | $p < 0.004$ ** | $p < 0.004$ ** | NS | $p < 0.004$ ** |
| 5000+ | NA | NA | NA | NS | NA | NS | NS |

**Table A7.** Coastal and central reserve rare species richness by vegetation community.

| Vegetation Community | Coastal Vegetation Acres | Coastal Rare Plant Species Richness | Central Rare Plant Species Richness | Central Vegetation Acres |
|---|---|---|---|---|
| Coastal sage scrub | 12,789.52 | 36 | 30 | 17,167.85 |
| Non-native grassland | 4883.17 | 19 | 19 | 5305.378 |
| Riparian | 1786.568 | 19 | 16 | 1708.155 |
| Chaparral | 1970.494 | 18 | 21 | 8715.643 |
| Maritime chaparral | 1437.923 | 14 | 13 | 2052.204 |
| Saltmarsh | 688.9108 | 15 | 0 | 0 |
| Woodland | 997.0518 | 7 | 20 | 3055.707 |
| Dune | 95.78647 | 5 | 0 | 0 |
| Wash scrub | 0 | 0 | 3 | 126.1679 |
| Native grassland | 59.28799 | 4 | 7 | 686.0134 |
| Freshwater marsh | 161.2096 | 2 | 0 | 22.68983 |
| Cliff, rocky outcrop | 28.25677 | 1 | 6 | 289.0837 |
| Wet meadow | 3.35519 | 1 | 0 | 0.524679 |
| Evergreen forest | 0 | 0 | 3 | 33.83678 |

**Table A8.** Results of chi-square analysis for likelihood of rare plant occurrences within each vegetation community with bonferroni post hoc significance values listed for each category. *p* values with one asterisk (*) indicate that the number of rare species observations were significantly higher than expected (assuming random distribution across the reserve) two asterisks (**) indicate that the number of rare plant occurrences were significantly lower than expected. NS indicates insignificant results.

| Vegetation Community | Bonferroni Post Hoc Significance Value |
|---|---|
| Chaparral | NS |
| Cliff, Rocky outcrop | $p < 0.003$ ** |
| CSS | NS |
| Dune | $p < 0.003$ * |
| Evergreen Forest | $p < 0.003$ ** |
| Freshwater Marsh | $p < 0.003$ ** |
| Maritime Chaparral | NS |
| Native grassland | NS |
| Non-native grassland | NS |
| Riparian | NS |
| Salt Marsh | NS |
| Wet meadow | $p < 0.003$ ** |
| Woodland | NS |

$n = 4425$, Degrees of Freedom = 13, $\chi^2 = 2057.56$, *p* Value < 0.0001.

## Appendix C

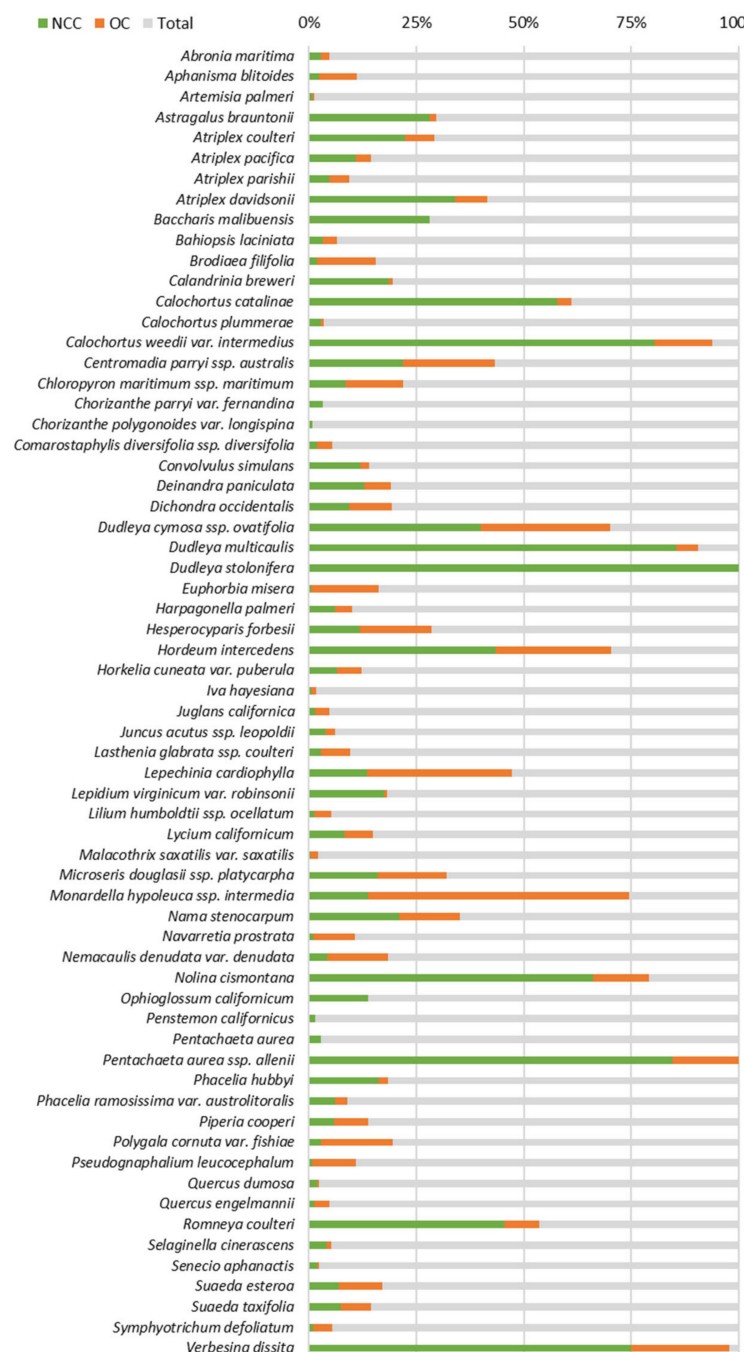

**Figure A1.** Percent of rare plant occurrences within the Natural Communities Coalition (NCC) Reserve System and Orange County relative to the total number of occurrences throughout a species' range. All species in the NCC (green) are within Orange County (orange), with gray indicating species occurrences outside of NCC and Orange County.

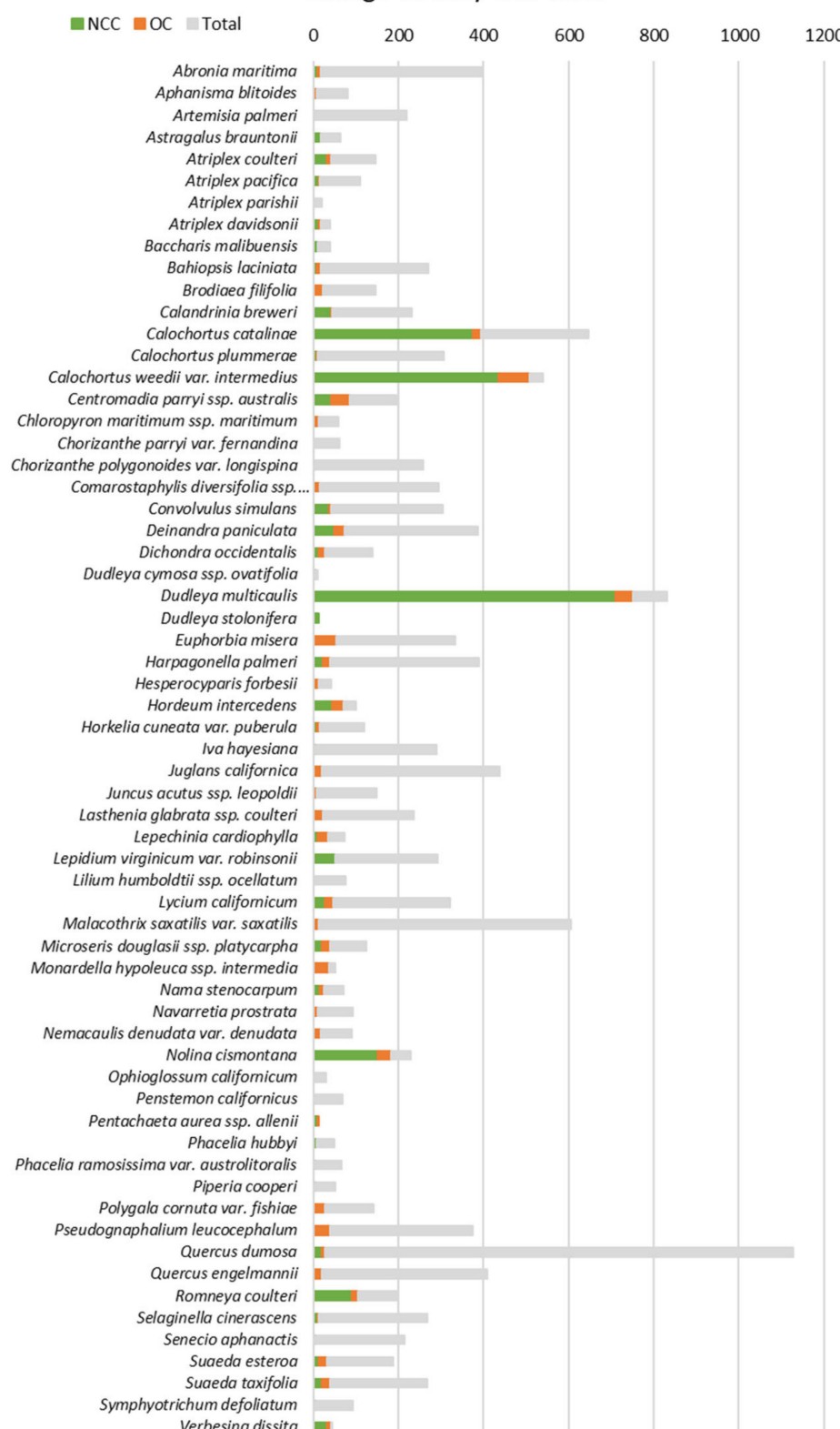

**Figure A2.** Occurrence numbers of the rare plant species of the NCC reserve within the NCC reserve (green), Orange County (orange), and each species total occurrences (gray).

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
