# Peer review of "Analysis of Rare Plant Occurrence Data for Monitoring Prioritization"

_diversity, doi:10.3390/d12110427_

Round 1
Reviewer 1 Report
This manuscript deals with an interesting topic of prioritizing rare vascular plant species for conservation from an environmental important part of western North America. However, there is a need to an early inclusion of the description of the study area for a broader audience (like in Table B3). The discussion on edge and vegetation communities could have mentioned the characteristics of patchiness of those communities, and if patchiness might have a role in the conservation threats for those 64 rare taxa.
There should be a mentioning of disturbance types, and as for the case of fire periodicity how it affects part or the whole life cycle (seed bank?) of these rare species. The bias of occurrence regarding disturbance sensitive taxa is a promising subject.
Line 89. Include between parenthesis area in hectares or square kilometers.
Line 98-99. Indication of 92 selected taxa as written indicates to be found in Table 2 in this manuscript. If the selected taxa were chosen from Table 2 of the CNPS list there should be a more specific citation. It needs to explain better steps described in lines 112 to 117 that resulted in the final number that was analyzed (see line 175 in results), and then re-write sentence in lines 163-164. It is only in lines 345 and 346 under Discussion that selection of species is better explained.
The Appendix has five tables, a sum of all tables under A, and the total indicated in Table B1 results in 64 taxa in total.
Table A1 includes a single lycopod, Selaginella cinerascens, but not included in B1, is this another species not analyzed besides listed Helianthus nuttallii ssp. parishii?
Table A3, change species names from capital letters to lower case. Verify spelling of all infraspecific taxa such as Lepidium virginicum var robinsonii (not “Cobinsonii”), Lilium humboldtii ssp. ocellatum (not “Ccellatum”), Pentachaeta aurea ssp. allenii (not “Cllenii”), and Polygala cornuta var. fishiae (not “Cishiae”)
Table B1, change capital letter to lower case for all subspecific taxa. Correct Chlorophytum maritimum ssp. maritimum (not “Aaritimum”), Malacothrix saxatilis var. saxatilis (not “Axatilis”). The legend includes the sunflower not analyzed but see comment above of A1. Instead there is the typical (?) Pentachaeta aurea, from A1
Line 152 One of the goals of this manuscript was to link rarity and vegetation communities, there should be a reference to the names, number, and area of those communities. These data are only found in Figure 4 and listed in Table 4.
Line 182 Redo sentence “The Coastal reserve has a higher rare plant species of 39 compared to 53…”, see Table 4.
Lines 235 and 238 are linked to Table 2. The legend needs to be changed from meters to kilometers.
Lines 252-254. Coastal reserve has low plant occurrence data. Give the number. If there are a total of 4490 occurrences, and the central part has the majority, then “there are 3000 fewer…” should be 1490 (?)
Lines 381 to 393. It is fire, the only disturbance regime or there are other disturbances as important.
Reviewer 2 Report
The evaluated manuscript fits in the scope of the Journal Diversity. It influences on the problem of rare plant species, its sampling and some difficulties related with its management in natural protected areas, which is a controversial and interesting topic.
The aims of the work are: the understanding of rare plants distribution in two disjunt protected natural areas from the state of California, USA (see lines 72-79), and also to further the consideration for rare plant species prioritization and monitoring efforts (lines 85-86).
The title, key words and abstract are appropriate and reflect the content of the evaluated article. Besides, the Introduction provides adequate information for understanding the article
About the Methodology, the authors create a database for the study area with the aim of analysing rare plant distribution and spatial patterns. In this way they calculated a rare plant species richness index and applied it to a grid system, in order to offer a cartography for visualizing and compare the two disjunt areas. Likewise they performed some different analyses in order to find the probability to find detect a rare plant in different locations. All methodological development seems correct.
The main drawback of the evaluated manuscript is related with the Results, Discussion and Conclusions sections. The results obtained, the discussion based on the aforementioned results and the conclusions, as they are written, have too short a horizon. They are aimed at too local a scope for this journal.
Taking into account the above considerations, if the authors want to publish this work in Diversity, they must rewrite these sections again, expressing clearly and evidently how their results contribute to increase the knowledge of plant rare species distribution and how they can make the management of those organisms, in natural protected areas, more effective, in relation to the results obtained. They should include examples and similar problems that have appeared in other places, in order to demonstrate the usefulness of the results obtained.
Round 2
Reviewer 2 Report
In the new version of the article, my recommendations have had little echo, in view of the revised text.
In relation to my suggestions, the text remains practically the same as the previous version (lines 351, 391 and 421). This is still a discussion of little interest to botanists, ecologists and managers of protected natural areas other than the territory where the study area is located.
I cannot find the reference they say they have added in line 368.
As I pointed out, the authors should include some examples and references to similar problems that have appeared in other places, in order to demonstrate the usefulness of the results obtained and to make the reader, who is situated in other places than the area of study, find the article, at least a little interesting.
